# Minimally Invasive Surgery in Liver Transplantation: From Living Liver Donation to Graft Implantation

Eleni Avramidou *, Konstantinos Terlemes, Afroditi Lymperopoulou, Georgios Katsanos, Nikolaos Antoniadis, Athanasios Kofinas, Stella Vasileiadou, Konstantina-Eleni Karakasi and Georgios Tsoulfas

Department of Transplant Surgery, Center for Research and Innovation in Solid Organ Transplantation Aristotle University of Thessaloniki School of Medicine, 54642 Thessaloniki, Greece; terlemesk@gmail.com (K.T.); alymperop@auth.gr (A.L.); katsanosg@auth.gr (G.K.); nikanton@auth.gr (N.A.); kofinasthanasis@gmail.com (A.K.); stvasileiadou@gmail.com (S.V.); ke_karakasi@hotmail.com (K.-E.K.); tsoulfasg@auth.gr (G.T.)
* Correspondence: avramidoue@auth.gr

**Abstract:** Since the end of the 20th century and the establishment of minimally invasive techniques, they have become the preferred operative method by many surgeons. These techniques were applied to liver surgery for the first time in 1991, while as far as transplantation is concerned their application was limited to the living donor procedure. We performed a review of the literature by searching in Pubmed and Scopus using the following keywords: Liver transplantation, Minimally invasive surgery(MIS) living liver donor surgery. Applications of MIS are recorded in surgeries involving the donor and the recipient. Regarding the recipient surgeries, the reports are limited to 25 patients, including combinations of laparoscopic, robotic and open techniques, while in the living donor surgery, the reports are much more numerous and with larger series of patients. Shorter hospitalization times and less blood loss are recorded, especially in centers with experience in a large number of cases. Regarding the living donor surgery, MIS follows the same principles as a conventional hepatectomy and is already the method of choice in many specialized centers. Regarding the recipient surgery, significant questions arise mainly concerning the safe handling of the liver graft.

**Keywords:** MIS; liver transplantation; donor surgery; recipient surgery; living donor liver transplantation; robotic hepatectomy; liver transplantation

## 1. Introduction

Since the beginning of minimally invasive and laparoscopic surgeries in the late 20th century, many surgeons use this method as their preferred approach regarding hepatectomy [1].The term of minimally invasive surgery (MIS) refers to the usage of smaller incisions for access, as an alternative to the regular open technique [2]. MIS techniques offer multiple advantages including less postoperative pain and better cosmetic results [3]. MIS is applicable to the majority of surgical procedures, including complex operations and some core ones, like cholecystectomy and appendectomy [4].

Today, MIS has evolved from the field of experimental medicine to daily clinical practice. Contraindications that used to exist were complexity of the surgery and existence of incisions in the abdominal area. Nowadays, the efficacy and safety of MIS, even on patients with the above contraindications, has been scientifically proven [5]. The above findings have led to the application of MIS in liver transplant recipients as well as living liver donors, with potentially better outcomes than the traditional open techniques [6,7].

MIS is a surgical approach option in liver surgeries since 1991, with the first laparoscopic resection reported by H. Reich [8]. Since that time and with the recent technological developments, MIS liver surgeries has increased [9]. The most common MIS liver surgeries include local non-anatomical resection and left lateral segmentectomy, with procedures like

isolated caudate lobectomy, trisegmentectomy and middle hepatic lobectomy (segments 4, 5, and 8) being more rare but still a feasible option [10–12].

According to the above, MIS can be an applicable approach in many surgeries related to liver transplantation (LT). MIS techniques in liver transplantation were first used in 2002 in living donor liver donation, when Cherqui et al. first reported resection of left lobe liver graft with laparoscopy [13]. In the past, application of MIS techniques has been a matter of debate due to the fear of graft loss. During recent years, MIS application reports in LT recipients have increased, with the majority of them relating to the treatment of postoperative applications [7].

To the best of our knowledge, there is no review summarizing the current status of minimal invasive approach in LT patients and donors. Thus, the present narrative review was conducted to assess the minimally invasive approach in this cohort of patients with focus on living donor surgery and liver transplant recipient surgery.

## 2. Materials and Methods

We conducted a literature review of the medical research databases PubMed and Scopus. We used the following key words for our search: Liver transplantation, Minimally Invasive surgery, living donor hepatectomy and recipient surgery. Our research was limited to the period from 2000 to the present and includes studies written in the English language. Our exclusion criteria included articles in languages other than English, studies that were not human-related, and bibliographies that did nor refer to application of MIS techniques in LT.

## 3. Results

Applications of MIS in living donor surgeries can be found in Tables 1 and 2, while applications of MIS regarding recipient surgery can be found in Table 3.

### 3.1. Minimally Invasive Surgical Techniques in Living Donor LT

3.1.1. Laparoscopic Donor Hepatectomy

Cherqui et al. in 2002 were the first to perform laparoscopic left hepatic lobectomy in two donors for living donor liver transplantation. Operations lasted 6–7 h, with minimal blood loss and no complications [13]. Regarding adult living donor LT, pure laparoscopic donor left hepatectomy was first reported eleven years later, in 2013 [14]. The successful surgeries demonstrated the potential for laparoscopic donor left lobectomy as a safe and effective method for pediatric and adult living donor liver transplantation, signaling a significant advancement. In recent years, laparoscopic major hepatectomy has been standard practice among expert teams. In a laparoscopy-assisted donor right hepatectomy (LADRH), the donor is placed in the supine position with arms abducted. If the hand assisted technique is chosen, a midline subxiphoid incision measuring 8 cm is created to provide hand assistance during the mobilization of the liver and the extraction of the graft. After using a 5-mm umbilical camera port to view the liver, an additional 5-mm port is inserted either at the right flank or through the Gel Port. The surgeon, positioned on the right side of the donor, employs hook-type electrocautery through the right flank port to dissect and divide the ligaments, continuing the mobilization process until reaching the lateral aspect of the inferior vena cava [15]. Regarding the Left hepatectomy (LALDH), the donor position and settings of the laparoscopic procedure were the same as the right hemihepatectomy [16,17]. In the Pure Laparoscopic Left Lateral Sectionectomy (PLLLDS), five trocars are inserted: three of 12-mm diameter and two of 5-mm diameter and a 30-degree laparoscope is employed for visual assessment of the liver. The mobilization of the left lateral section is accomplished by cutting the round, falciform, and left triangular ligaments using a harmonic scalpel [18]. Regarding operative time in LALDH, it ranged from 265 min to 702.50, with different operation times being reported between left lateral and left hemihepatectomy, as well as differences occurring in different centers based on the surgical experience of each [14,19–32]. Additionally, Takahara et al. and Seong

et al.found that operative time in LALDH was shorter than the pure laparoscopic left donor hepatectomy (PLLDH) [26,30]. As for the right hepatectomy, operative time ranged from 181.0 min to 1065 min. [26,31,33–50], with the differences being reported caused mainly by the surgical experience of each center.

Post Operative Outcome in Laparoscopy Assisted Donor Hepatectomy

Hospital stay of donors ranged from 4 days to 50 days, with shorter hospital stay being reported in LLS [31,39,42]. Min et al. and Song et al. reported a slightly shorter hospital stay for LADH group than the PLDH group [21,30]. Concerning blood loss, this parameter ranged from 10 mL to 1559 mL, with the minimal blood loss being reported by Scarton et al. in LLLS and maximum blood loss being reported in a case of hybrid laparoscopic donor hepatectomy [18,27].

Complications were evaluated using the Clavien-Dindo Classification (I–IV). Major complications (grade III and above) were reported by Song et al. and included post-operative ileus, hemorrhage which required laparotomy and post hepatectomy liver failure [30]. Kitajima et al. also reported three bile leakage cases requiring endoscopic nasobiliary drainage in left liver hepatectomy living donors [27]. Safwan et al. reported two major complications that included one patient that developed postoperative bleeding which required re-laparotomy and one patient that required video-assisted thoracoscopic surgery for the management of loculated pleural effusion [23]. Choi et al. in their series of patients reported pleural effusion, biliary stricture and diaphragmatic hernia [29]. Makki et al. also reported eight cases of donors with major complications requiring intervention [28]. T. Kobayashi et al. reported only one major complication occurring in LLLS [27]. Lastly, Marubashi et al. reported two patients with delayed gastric emptying which required fiberoptic endoscopy for correcting rotation of the stomach, and both recovered within two weeks after the donor surgery [28]. Regarding the postoperative pain, this is reported to be less than in the conventional open technique [51].

3.1.2. Robotic Assisted (RA) Donor Hepatectomy

In recent years many experienced centers and surgeons published their series of surgeries including all types of hepatectomy. The first ever right lobe robotic living donor hepatectomy was performed in 2011 by Guilianotti at the University of Chicago-Illinois, who already had years of experience in robotic MIS [52]. Chen et al. were the first to publish a series of robotic right lobe donor hepatectomy in 13 donors and compared them with 54 open living donor hepatectomy cases [53]. When it comes to robotic Left lateral sectionectomy, Liao et al. were the first to attempt it [54]. Four years later, Troisi et al. were the first to perform a series of 25 robotic left lateral sectionectomies and compare them with 50 laparoscopic left lateral sectionectomies, completing the series of patients with zero conversions with the robotic approach compared to two with the laparoscopic [55]. In the last couple of years Robotic living donor hepatectomy (RLDH) is starting to gain attention internationally with a few other experienced centers publishing their results as you can see in Table 2 [56,57]. In 2020 a combination of Right lobectomy (RL), Left Lateral Sectionectomy (LLS), and Left lobectomy (LL) surgeries were performed by Broering et al. and 2 years later Schulze et al. published their series of 501 RLDH including RL, LL and LLS with very encouraging results, showing that experience with robotic surgery makes it possible to perform any kind of robotic living donor hepatectomy regardless of anatomical variations and graft size [58,59]. The operation time in those series of surgeries ranged between 290–596 min. Naranjo et al. found that RLDH lasted on average 133.4 min longer than OLDH and 137.7 min longer than LLDH [60].

Post Operative Outcome in RA LDH

The intraoperative blood loss ranged from 50–1000 mL, but in general it was very similar to OLDH (open living donor hepatectomy) and slightly less than LADH (laparoscopy assisted donor hepatectomy) [58,61]. All studies comparing the length of postoperative

hospital stay between ODH and RLDH found it to be lower in RLDH. Rho et al.were the only ones not to find any differences in hospital stay between LADH and RLDH [61] In a systematic review/meta-analysis comparing RLDRH and ODRH, Naranjo et al. found no differences in postoperative mean peak ALT and AST levels respectively, but the postoperative mean peak total bilirubin level was lower in RLDRH. When they compared the Robotic living donor right hepatectomy (RLDRH) versus the Laparoscopic assisted donor right hepatectomy (LADRH) they showed that the postoperative mean peak ALT and AST levels were higher in RLDRH. Conversely, it was shown that there was no difference in the postoperative mean peak total bilirubin level. [60]. In the same systematic review, when comparing the postoperative pain of RLDH vs. OLDH (>day 3) they found no differences, but when comparing RLDH to LLDH, the pain score after day 3 was lower in RLDH. [60]. Complications were evaluated using the Clavien-Dindo Classification (I–IV) [62–64]. Most common complications included pleural effusion, biliary leak and hepatic artery bleeding and thrombosis.

### 3.2. Minimally Invasive Surgical Techniques in LT Recipient

Even though MIS techniques are widely used in Hepato-pancreato-biliary (HPB) surgery the last several years, their application in liver transplant recipient surgery is minimized to 25 reported patients, with 8 of them requiring conversion to open surgery [15,65–68]. Eguchi et al. were the first to describe the possible applications of MIS in LT recipient surgery [69]. Particularly they applied a hand-assisted laparoscopic surgery approach in 9 LT recipients though the explant of the diseased liver and then they converted the surgery to open through an upper midline incision. Even though they mentioned higher warm ischemia times compared to the conventional open technique, their technique showed no further limitations and proposed advantages mainly concerning the smaller incision that was applied. The following ten years no reported attempt was made concerning the introduction of MIS in LT recipients. Dokmak et al. were the first that attempted pure laparoscopic total hepatectomy in LT recipients, applying a hybrid LT surgery [65]. The following year, in 2021, Suh et al. were the first to successfully proceed to a pure laparoscopic LT both for the hepatectomy and for the graft implantation in the recipient [65,66]. Moreover, Lee et al. were the first to introduce the robotic assisted system in recipient LT surgery, by applying a hybrid laparoscopic and robotic LT approach [66]. Lastly, experimental MIS LT surgeries have been carried out successfully using experimental animals [70].

### 3.2.1. Explant Surgery

In all seven reports regarding MIS in LT recipient surgery the hepatectomy part of the liver transplantation was done laparoscopically. This surgical procedure follows the same principles as any partial hepatectomy, with some modifications in vessel clamping and division of vascular structures [71–73]. Mean surgical time ranged from 390–1220 min and warm ischemia time range was 30–117 min. In the beginning of the operation five trocars of various diameters are placed. In cases of a recipient with unresectable liver cancer, the surgeon observed the cavity for possible metastasis of the tumor [65]. Then, the procedure continues with cholecystectomy and lymphadenectomy in case of recipients with liver tumors. The hilar dissection includes the dissection of both branches of the hepatic artery, of the common bile duct, as well as the dissection of the portal vein based on the principles of pringle manoeuvre, a common approach in partial liver resection surgeries [74]. Additionally, the hepatectomy included the preparation of the recipient's right hepatic vein for the anastomosis with the grafts vessels [67]. It should also be mentioned that Dokman et al., were the first reported surgeons to succeed with the MIS LT approach. They facilitated the total hepatectomy by removing the liver with a two step procedure, by first applying a left lateral sectionectomy and then removing the rest of the liver [65].

### 3.2.2. Graft Implantation Surgery

After the total hepatectomy, the next step in the LT is the graft implantation. The MIS approach for graft implantation includes open surgery though an upper midline incision, laparoscopic surgery or even robotic assisted surgery purely or in combination with other techniques [65–69,75]. In the laparoscopic and robotic assisted surgeries, a gel port was frequently placed in the incision of the previous liver extraction, that could be used as a hand port in case of an emergency [66–70]. Additionally, a small incision was made in the left upper quadrant for the insertion of the Chitwood clamp for the suprahepatic IVC clamping as well as the implantation of the liver graft [66,67]. After the reperfusion, in some approaches the surgery switched from laparoscopic to robotic assisted [66]. The last step included the laparoscopic or robotic assisted en- to-end anastomosis of the bile duct [76].

### 3.2.3. Post Operative Outcome in LT Using MIS Technique

Median range of hospital stay was 11–28 days and blood loss ranged from 250–3600 mL. LT is a major procedure with many major or minor complications [76–83]. Some of the most common post-operative LT complications that require surgical intervention include biliary complications (biliary peritonitis, biliary leakage, biliary stricture), internal hernias, ascites, abscesses and rejection-related complications [84–88]. Interventional radiology procedures can be used for the management of some of these complications with great success [87]. In cases that require surgical intervention, there is the potential for a MIS approach in the treatment of post LT complications [87–92].

**Table 1.** Applications of Laparoscopic assisted surgery in living donor hepatectomy surgery.

| Authors | Date of Publication | Type of Study | N of Patients | Type of Surgery | Operative Time (min) | Blood Loss (mL) | Conversion Rate | Complications | Total Hospital Stay (Days) |
|---|---|---|---|---|---|---|---|---|---|
| Moon et al. [36] | 2022 | Retrospective study | 3 | Right hepatectomy | 565–750 | 200–300 | 0/3 | Bile leakage (1) | 9–15 |
| SuhKyun et al. [15] | 2022 | Comparative study | 213 | Righ hepatectomy | 289.9 ± 54.9 | 306.1 ± 213.1 | NA | Wound problem (3), Pulmonary thromboembolism (1), biliary problem (20), Intra-abdominal fluid collection (1), portal vein thrombus (1), Pleural effusion (1), bleeding (1) | 20.6 ± 15.4 |
| Hong et al. [31] | 2021 | Retrospective multicenter Study | 545 | Right Hepatectomy (481) Left Hepatectomy (25) LLS (39) | Right hepatectomy: 340.1 ± 106. Left hepatectomy 308.5–409 LLS 341.6 ± 66.2 | 302.5 Right hepatectomy 316.3 ± 233.7 Left hepatectomy 300.0 (150.0–400.0) LLS139.5 ± 117.2 | 10/545 | Wound problem (7), pleural effusion (13), intra-abdominal fluid collection (4), bile leakage (15), portal vein thrombosis (2), pulmonary thromboembolism (1), biliary stricture (3), portal vein stenosis (1), intra abdominal bleeding (4), shock (1) | 9.4 Right hepatectomy: 9.4 ± 3.6 Left hepatectomy: 7.0–10.0 LLS: 9.2 ± 2.8 |
| Han et al. [38] | 2021 | Comparative study | 100 (43 before the learning curve, 57 after the learning curve | Right hepatectomy | Before: 282.2 ± 59.2 After: 181.0 ± 35.7 | Before: 344.4 ± 224.0 After: 161.4 ± 130.0 | NA | Before:Intra-abdominal fluid collection (2) Biliary stricture (1), intra abdominal bleeding (1) After: Portal vein thrombosis (1) | Before: 7.1 ± 2.4 After: 5.8 ± 1.4 |
| Han et al. [44] | 2021 | retrospective case series. | 300 donors divided into three subgroups of periods 1–3 of 100 cases each: 1–100, 101–200, and 201–300 | Right hepatectomy | 267.8 ± 74.2 | 261.5 ± 209.8 | NA | Wound problem (3), pleural effusion (1), intra-abdominal fluid collection (6), portal vein thrombosis (1), pulmonary thromboembolism (1), portal vein stenosis (2), bilary problems (4), bleeding (1) | 7.4 ± 2.6 |
| Cho et al. [45] | 2021 | Comparative study | 90 | Right hepatectomy | 364 | 175 | 0/90 | None | 8.2 |
| Seon Jeong et al. [34] | 2020 | Report | 123 | Right hepatectomy (119) Extenede right hepatectomy (4) | 335 ± 95 | 300 | | Pleural effusion (29), Atelectasia (9), Bile leakage (7), bile duct stricture (3), bleeding (1), portal vein narrowing (1), fluid collection (2), wound complications (7) | 9 (8–11) |

**Table 1.** *Cont.*

| Authors | Date of Publication | Type of Study | N of Patients | Type of Surgery | Operative Time (min) | Blood Loss (mL) | Conversion Rate | Complications | Total Hospital Stay (Days) |
|---|---|---|---|---|---|---|---|---|---|
| Lee et al. [39] | 2019 | Scientific report | 35 | Right hepatectomy | 433.7± 142.9 | 572.2 ± 438.9 | 2/35 | Wound problems (3), portal vein thrombosis (1), portal vein stricture (1), bleeding (1) | 9.70 ± 4.35 |
| Rhu et al. [47] | 2019 | Comparative study | 100 | Right hepatectomy | 375.2 ± 94.0 | 299.3 ± 161.7 | 6/100 | Wound problem (1), ileus (3), fluic collection (1), biliary complication (8), bleeding (1). | 11.0 ± 4.0 |
| Park J et al. [40] | 2019 | Case-control study | 91 | Right hepatectomy | 345 | 300 | NA | Wound problems (3), bile leakage (11), fluid collection (1), biliary complications (1), vascular complications (2) | 10 |
| Song et al. [32] | 2019 | Case report | 1 | left hepatectomy. | 495 | <100 | NA | None | 6 |
| Kwon et al. [33] | 2018 | Retrospective cohort study | 54 | Right without MHV (41) Extended right with MHV (10) Left with MHV (3) | 436 (294–684) | 300 (10–850) | 4/54 | Wound infection (3), ileus (2), bile leakage (10) portal vein stenosis (3), bleeding (1) | 10 (7–27) |
| Broering et al. [51] | 2018 | Observational study | 72 | LLS | | 100 (50–600) mL | 3/72 | Bile leakage (2) | 4.1 ± 1.33 |
| Lee et al. [49] | 2018 | Retrospective study | 115 | Right hepatectomy | 321.5 ± 57.2 min | 394.1 ± 197.6 | NA | Wound problems (2), pleural effusion (1), intra-abdominal fluid collection (2), portal vein thrombosis (1), bile leakage (1), biliary stricture (1), bleeding (1) | 7.8 ± 1.8 |
| Hong et al. [37] | 2018 | Retrospective study | 26 | Right hepatectomy | 304.5 (58.7) | | 0/26 | Intra-abdominal fluid collection (1), bleeding (1), hepatic xiphoid trocar injury (2) | 7.7 ± 3.0 |
| Song et al. [30] | 2018 | Comparative study | 7 PLRH 26 HARH | Right hepatectomy | PLRH: 509.3 ± 98.9. HARH: 451.6 ± 89.7 | PLRH: 378.6 ± 177.1 HARH: 617.3 ± 240.4 | | PLRH: Pleural effusion (1), infection (1) HARH: Bile leakage (2), ileus (1), infection (1), hemorrhage (1), Liver failure (1) | PLRH: 7.7–10 HARH: 7.5–12 (8.5) |
| Safwan et al. [23] | 2018 | Retrospective Comparative study | 19 | Hybrid Right hepatectomy | 375.5 ± 51.9 | 228.9 ± 123.1 | NA | Ileus (2), deep vein thrombosis (1), Thrombophlebitis (1), fluid collection (1), bile leakage (1) | NA |

**Table 1.** *Cont.*

| Authors | Date of Publication | Type of Study | N of Patients | Type of Surgery | Operative Time (min) | Blood Loss (mL) | Conversion Rate | Complications | Total Hospital Stay (Days) |
|---|---|---|---|---|---|---|---|---|---|
| Suh et al. [43] | 2018 | Comparative study | 45 | Right hepatectomy | 330.7 ± 49.5 min | 436.0 ± 170.3 | NA | Intra abdominal bleeding (5), intra abdominal fluid collection (4), wound problems (5), hepatic artery problems (4), portal vein or hepatic vein problems (2), biliary problems (12) other complications (9) | PLRDH: 8.2 ± 1.3 days |
| Kobayashi et al. [20] | 2018 | Retrospective study | 11 | Graft type (right lobe/left lobe/posterior section/left lateral section) LAP ASSISTED 4/5/1/1 | 475 (400–645) | 350 (15–1128) | NA | biliary fistula (1) | 10 (7–19) days |
| Eguchi et al. [69] | 2018 | Comparative study | 110 | right hemihepatectomy: 43, extended left Hemihepatectomy: 66, and right lateral sectionectomy: 1 | 405 (286–671) | 537 | NA | Wound problem (1), bile leak (3), ileus (2), bleeding (2), portal vein thrombosis (1) | 13 (6–40) |
| Rotellar et al. [41] | 2017 | Comparative study | 5 | Right hepatectomy | 476 (420–480) | <200 | 0/5 | Infection (2) | 4 (3–5) |
| Kim et al. [56] | 2017 | Case reports | 3 | LDRH | 447–502 | 200–270 | 0/3 | NA | 7–8 |
| Kitajima et al. [27] | 2017 | Observational study | 76 | Right hepatectomy: 41 Left hepatectomy: 35 | Right: 431 (310–651) Left: 459 (310–633) | Right: 201 (10–1559) Left: 245 (22–1840), no transfusions | 5/76 | NA | Right: 12 (8–27) Left: 12 (7–50) |
| Takahara et al. [26] | 2017 | Comparative study | 40:LADH 14: PLDH | Left and right hepatectomies | LADH: 380.40 ± 44.08. PLDH: 454.93 ± 85.60 | PLDH 81.07 ± 52.78 LADH 238.50 ± 177.05 | | PLDH: biliary complications (3) LADH: biliary complications (4).wound infections (2) other complication (1) | LADH 9.05 ± 3.30 PLDH 8.43 ± 1.65 |
| Scatton. et al. [18] | 2015 | Prospective cohort study | 70 | 67 donors underwent LLS, and 3 underwent LH without middle hepatic vein procurement. | 175–520 | 10–770 | 4/70 | biliary leakage (2), Biliary stenosis (1) Pulmonary complications (2) Pneumothorax (1), Respiratory infection (1) Bladder injury (1) Wound complications (5) Infection (1) Hematoma (4) Gastric ulcer (1) | hospital stay 3–18 |

**Table 1.** *Cont.*

| Authors | Date of Publication | Type of Study | N of Patients | Type of Surgery | Operative Time (min) | Blood Loss (mL) | Conversion Rate | Complications | Total Hospital Stay (Days) |
|---|---|---|---|---|---|---|---|---|---|
| Suh et al. [24] | 2015 | Comparative study | 14 | Right hepatectomy | 333.8 ± 61.7 | 298.3 ± 118.8 | NA | None | 8.4 ± 1.6 |
| Han et al. [48] | 2015 | Case report | 2 | Right Hepatectomy | NA | NA | NA | None | 10 8 |
| Makki et al. [28] | 2014 | Observational study | 26 | Right hepatectomy | 336.54 ± 89.40 | 702.50 ± 124.11 | NA | Wound problems (3), pleural effusion (1) | NA |
| Zhang et al. [22] | 2014 | Prospective case matched study | 25 | Right hepatectomy | 385.9 ± 47.4 | 378.4 ± 112.5 | NA | Pleural effusion (1), pulmonary infection (2), bleeding (1), | 7.0 ± 1.4 |
| Soubrane et al. [42] | 2013 | Case report | 1 | Right hepatectomy | 480 | 100 | NA | none | 7 |
| Samstein et al. [14] | 2013 | Case report | 2 | left Hepatectomy | 358 and 379 | 125 | 0/2 | Bile leakage (1) | 5-3 |
| Marubashi et al. [19] | 2013 | Retrospective comparative study | 31 | Left hepatectomy | 435 ± 103 | 353 ± 39.6 | NA | NA | 10.3 ± 3.3 |
| Choi et al. [29] | 2012 | Retrospective comparative study | 40:SPLADRH 20: LADRH | Right hepatectomy | Single port 278.50 ± 72.25  Laparoscopy assisted 383.55 ± 41.73 | Singles port 450.0 ± 316.43  Laparoscopy assisted 870.0 ± 653.01 | SPLADRH: 2/40 LADRH: 2/20 | SPLADRH: pleural effusion (1), bile leakage (3), bleeding (2) LADRH: wound complication (2), pleural effusion (2), billary stricture (1) | Single port 11.8 ± 4.45  Laparoscopy assisted 12.1 ± 2.81 |
| Baker et al. [25] | 2009 | Comparative study | 33 | Right hepatectomy | 265 ± 48 | 417 ± 217 | 2/33 | NA | 4.3 |
| Soubrane et al. [93] | 2006 | Retrospective comparative study | 16 | Left lateral sectionectomy | 320 ± 67 min | 18.7 ± 44.2 mL | None | wound hematomas (2) 1bile leak (1) | 7.5 ± 2.3 days |
| Koffron et al. [94] | 2006 | Case report | 1 | Right hepatectomy | 235 | 150 | NA | NA | 3 |
| Seog et al. [21] | 2006 | Comparative study | 20 | LADH: LLS 7 Left: 1 Right: Y 1  PLDH: LLS 4 Left 6 | Lap-assisted 351.0 ± 137.7  Pure Lap  458 ± 123.0 | NA | 1/10 Lap assisted | Lap-Assist group: atelectasis (2) bile leakage (1) | Lap Assisted 16.4  Pure lap 11.5 |
| Cherqui et al. [13] | 2002 | Case series | 2 | Left hepatic lobectomy | 360 420 | 150 and 450 | NA | None- | 7 5 |

NA: Not applicable, MHV: Main hepatic vein, LLS: Left lateral sectionectomy, HARH: Hand assisted right hepatectomy, LH: Left hepatectomy, RH: Right hepatectomy, PLDH: Pure laparoscopic donor hepatectomy, LADH: Laparoscopy assisted donor hepatectomy, SLADRH: Single port laparoscopy assisted donor right hepatectomy, LADRH: Laparoscopy assisted donor right hepatectomy.

**Table 2.** Applications of Robotic Surgery in living donor surgery.

| Authors | Date of Publication | Type of Study | N of Patients | Type of Hepatectomy | Operative Time (min) | Average Blood Loss (mL) | Convention Rate | Complications | Length of Hospital Stay (Days) |
|---|---|---|---|---|---|---|---|---|---|
| Kim et al. [56] | 2022 | Retrospective cohort study | 102 | RL | 464 | 104 | NA | NA | 8.7 ± 3.1 |
| Schulze et al. [59] | 2022 | Cohort study | 501 | RL, LL, LLS | 406.2 | 60 (20–800) | 2/501 | Abdominal fluid collection (3), beeding (2), bile leakage (9), deep vein thrombosis (2), hematoma (12), pulmonary embolism (3) | 4 (2–22) |
| Jeong Jang et al. [57] | 2022 | Case series | 10 | RL | 396.6 ± 62.7 | NA | 0/10 | None | 8.7 ± 2.6 |
| Rho et al. 2022 [61] | 2022 | Comparative study | 52 | RL | 493.6 ± 91.5 | 109.8 ± 101.5 | NA | Minor complications (10), pleural effusion (1) hepatic artery bleeding (1) | 9 ± 2.1 |
| Troisi et al. [55] | 2021 | Retrospective comparative study | 25 | LLS | 290 ± 45 | 50 (30–250) | 0/25 | None | 3 (2–5) |
| Broering et al. [58] | 2020 | Single Center review | 175 | 80 RL, 34 LL, 61 LLS | 424 (177–693) | 138.1 (20–1000) | NA | biliary leak (3) | 4.3 (2–22) |
| Broering et al. [51] | 2020 | Comparative study using propensity score matching. | 35 | RL | 504 ± 73.5 | 250 (100–800) | NA | Minor complications (2), biliary leak (1), pulmonary embolism (1) | 5 (3–12) |
| Liao et al. [54] | 2017 | Case Report | 1 | LLS | 390 | 400 | NA | None | 8 |
| Chen et al. [53] | 2016 | Case Series | 13 | RL | 596 (353–753) | 169 (50–500) | NA | Hepatic artery thrombosis (1), biliary complications (1) | 7.0 (6–8) |
| Guilianotti et al. [52] | 2011 | Case Report | 1 | RL | 460 | 350 | NA | Late portal vein stenosis | 5 |

NA: Non applicable, RL: Right lobectomy, LL: Left lobectomy, LLS: Left lateral sectionectomy.

**Table 3.** Applications of MIS in liver transplant recipients.

| Authors | Date of Publication | Type of Study | N of Patients | Type of Surgery | Conversion Rate | Anhepatic Phase (min) | Ischemia (min) | Operative Time (min) | Blood Loss (mL) | Complications | Total Hospital Stay (Days) |
|---|---|---|---|---|---|---|---|---|---|---|---|
| Kim et al. [67] | 2023 | Case series | 10 | Laparoscopic (explant) Open (implant) | 6/10 | 48–152 | Cold: 105–234 Warm: 23–117 | 400–840 | 600–24.200 | Bleeding | 14 |
| Dokmak et al. [68] | 2022 | Case series | 6 | Laparoscopic (explant) Open (implant) | NA | 40–67 min | cold: 360–575 warm: 30–40 | 390–450 | 250–600 | NA | 10–14 |
| Suh et al. [48] | 2022 | Case report | 1 | Laparoscopic (explant) Laparoscopic-robotic (implant) | No | NA | NA | 1065 | 500 | Bleeding at the site of the suprapubic incision | 13 |
| Lee et al. [66] | 2022 | Case report | 1 | Laparoscopic (explant) Robotic (implant) | No | NA | Warm: 87 Cold: 220 | 1220 | 3600 | Mild reperfusion syndrome, Renal dysfunction, mild periportal edema, early graft dysfunction and postoperative ascites | 19 |
| Suh et al. [15] | 2021 | Case series | 5 | Laparoscopic (explant) Open (implant) | 2/5 | 25–201 min | Warm: 27–56 min | 499–640 min | 1750–7800 ml | bile leakage (2) and bleeding (1) | 15–30 |
| Dokmak et al. [65] | 2020 | Case report | 1 | Laparoscopic (explant) Open (implant) | No | 43 min | Warm: 38 min Cold: 466 min | 400 min | 400 mL | Mild ischemia reperfusion syndrome | 15 |

NA: Not applicable.

## 4. Discussion

To the best of our knowledge, this is the first review in the literature addressing the minimally invasive approach in the context of LT, both in donors and in recipients. Compared to the applications of MIS in other fields of surgery, there has been a delay in the incorporation of minimally invasive approaches in LT particularly in recipients, mainly fearing the possible damage or loss of the grafts. All of the MIS applications in LT recipient surgeries are dated after 2020 and include total MIS, combination of MIS and open techniques and laparoscopic and robotic surgical approaches.

The main concern and reason of the small number of LT recipients being operated with MIS is the maintenance of safety of the patient and the graft. The most common complications that lead to the conversion of the surgery include massive haemorrhage and hemodynamic instability of the patient. For this reason, a practical solution could be the utilization of hand assisted technique for the facilitation of emergency interventions by the surgeon, as well as the application of a Chitwood clamp and multiple laparoscopic bulldog clamps [46]. Moreover, based on the docking that the robotic surgery requires, a safer more time efficient approach for the implantation part of the surgery would be the one of laparoscopic surgery [46]. Additionally, MIS application in recipient surgeries seems to be more suitable for smaller liver grafts [15].

Apart from the main safety concerns regarding MIS applications in LT, another main point of consideration is the long and difficult learning curve of MIS applications in these complex surgeries. It is reported that the learning curve for laparoscopic surgeries is steep, in contrast with the simpler linear curve of the open techniques [94,95]. Moreover, according to recent European guidelines, training in laparoscopic liver surgeries should follow a stepwise progression from open to laparoscopic and then robotic surgical techniques, which due to the limited number and specialized nature of these procedures, training in MIS should be part of fellowship programs [96,97]. Another point of consideration is that up to today, very few liver transplant centers apply MIS techniques in everyday clinical practice, limiting training possibilities. These obstacles can be overcome with the usage of various simulation and VR applications. It has been reported that the application of technology in surgical education has positive outcomes leading to less complications and shorter operative time, even in complex surgeries [98–104].

Another reason for the limited application of MIS techniques in everyday clinical practice in liver transplantation is the financial cost. Generally, surgeries applying MIS techniques cost more than the open surgery alternative, with cost increases sometimes reaching even 100% [105–108]. The additional cost includes the equipment needed for those surgeries as well as the training of the surgeon and stuff (nurses, technicians etc.). Particularly, the cost of the equipment is the main reason that explains the cost differences between different MIS techniques and specifically between laparoscopic and robotic surgeries, with robotic assisted surgery equipment costing more [105].

Regardless of the surgical technique applied, in LT and LLD surgeries postoperative complications such as hernias and bile duct injuries can occur. MIS techniques have been applied in treating those complications, even in LT recipients that have a previous abdomen surgery. Most of the surgical interventions treat early postoperative complications, occurring in the first months after LT, particularly from day 5 till 8 months post operation [90–92]. Usage of robotic assisted surgery has been reported for the successful management of late anastomotic biliary stricture even 2 years after transplantation [88]. Incisional hernias are a very common post operative complication in open LT [109–120]. Unlike others, MIS approach for incisional hernia repair has been well documented regardless of when it occurred [111–120].

Despite the many difficulties MIS applications in LT surgeries may pose, there are several advantages for both the donor and the recipient. Concerning the recipient LT surgery, among the main advantages of the minimally invasive approach is the control of tissue damage as a trigger of the innate systemic inflammatory response, i.e., fewer acute phase reactants, lower CRP and complement, synthesis and activation of macrophages, and

natural killer and endothelial cells [121]. Moreover, there is a reduction in hospital stay and thus a lower risk of infectious complications [122]. The above advantages in combination with the decreased possibility of wound infection due to smaller incisions can prove to be very beneficial in transplant recipients who receive immunosuppression [123]. MIS have significant advantages for the donor surgery as well, mainly including better cosmetic outcome, shorter hospital stay and less perioperative pain, all of these being factors which apart from the well-being of the donor, are also bound to increase donation rates [124,125].

MIS applications in LT surgeries are still very limited with no official evidence-based guidelines yet. Rho et al. are the only ones reporting some indications regarding applications of robotic assisted surgery techniques in living donor liver surgery [61]. With the known superiority of minimally invasive techniques in obese patients, it is right to suggest the evidence based application of those techniques in LLD surgeries of obese donors, increasing this way the number of possible living donor liver donations [126]. Generally, it is reported that centers with experience and a higher volume of robotic surgeries managed to achieve shorter operative times. On the other hand, centers with less exposure to such procedures had an extended length of surgical time.. Based on the existing literature, the decision of application of MIS in LT and LLD should be made based on the patient characteristics and wishes as well as the experience of the transplant centre and surgeon.

Because of the limited applications of MIS techniques in LT, it is difficult to define its limitations as well as its advantages. Our review also has some limitations. Some of them include the quality and the degree of evidence of the studies, as it is limited mostly to single-center case series studies. Prospective cohort and randomized control trial studies should be held in the future in order to systematically record the advantages and limitation of applications of these type of surgeries in LT.

### 5. Conclusions

This review summarizes the reports of MIS in the LT surgeries, from the liver living donor surgery to the treatment of long term postoperative complications. For the optimal systematic evaluation of the effect of these type of surgeries in LT, national and international registries of LT surgeries, for the better investigation of preoperative and long-term outcomes should be organized.

**Author Contributions:** E.A. conceptualized the idea; visualized, collected, and analyzed the data; and wrote the manuscript. K.T. provided resources, analyzed the data and wrote the manuscript; A.L. provided resources, analyzed the data and wrote the manuscript; G.K. critically reviewed the manuscript; N.A. critically reviewed the manuscript; A.K. critically reviewed the manuscript; K.-E.K. critically reviewed the manuscript; S.V. critically reviewed the manuscript and G.T. supervised, assisted with the data curation, and edited the manuscript. All authors have read and agreed to the published version of the manuscript.

**Funding:** This research received no external funding.

**Conflicts of Interest:** The authors declare no conflict of interest.

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
