# Peer review of "Minimally Invasive Surgery in Liver Transplantation: From Living Liver Donation to Graft Implantation"

_livers, doi:10.3390/livers4010009_

Round 1

Reviewer 1 Report

Comments and Suggestions for Authors

The manuscript concerns the placement of minimally invasive surgery (MIS) as the technique used in liver transplantation. Authors present in the detailed way already gathered data on this subject, based on worldwide clinical experience. What I miss is a more expressive conclusion, with some kind suggestions related to particular techniques of liver transplantation that should be particularly applied in certain situations. What kind of hypothetical algorithm could be included in this area?

Comments on the Quality of English Language

The improvement of used expressions and grammar issues should be done.

Reviewer 2 Report

Comments and Suggestions for Authors

This study is based on a retrospective data review. Have you conducted a prospective data assessment that is supportive of your results?

Others:

This review addresses minimal invasive techniques in liver transplantation patients. The conclusions will require support with verification with complementary data at other centers’

This review is based on a relatively small number of patients but still is an important study.

The review should serve as a stimulus to other centers to evaluate the efficacy found in this study.

A shortcoming of this study is the small number of patients and the retrospective assessment. What has happened prospectively.

I think the conclusions are consistent with the evidence and arguments presented. The authors are asking for national and international registries assessing these types of patients.

The Tables and Figures are not ideal but certainly facilitate the discussion.

Comments on the Quality of English Language

Your paper will be of interest to the entire spectrum of liver transplantation personnel. Both alcoholic and non-alcoholic steatohepatitis are becoming the most common causes of cirrhosis,hepatocellular carcinoma and liver failure worldwide. 

Reviewer 3 Report

Comments and Suggestions for Authors

The study by Avramidou et al. reviews the evolution of minimally invasive techniques in liver surgery since 1991, initially applied in living donor transplantation. The literature search focused on liver transplantation and living donor surgery, with MIS showing benefits like shorter hospitalization and less blood loss, especially in experienced centers. The study highlights MIS as the preferred method in living donor surgery, though concerns exist about safe handling of the liver graft in recipient surgeries. This is a relative novel review summarizing the use of MIS in LT, and would provide useful information for applying such techniques in LT. However, some revisions are suggested.

1. MIS in LT recipients is still challenging, and higher conversion rate is observed in these studies, Some discussions on technique difficulties and the future improvement on MIS in LT recipients are encouraged.

2. While the review mostly discussed the advantages of MIS, it is suggested to summarize the shortcomings of using MIS, for example, is there a concern of cost when MIS is used.

3. The authors performed a through literature search to include multiple papers on this topic, however, a meta-analysis is encouraged to objectively compare the difference of different MIS strategies and open surgery. 

4. MIS should be spelled out in the title, as MIS is not a common abbreviation for readers.

5. English language and format editing are necessary throughout the whole manuscript. 

Comments on the Quality of English Language

Moderate editing.
